# PODiff: Latent Diffusion in Proper Orthogonal Decomposition Space for Scientific Super-Resolution

**Onkar Jadhav** [1]   **Tim French** [2]   **Matthew Rayson** [1]   **Nicole L. Jones** [1]

## Abstract

Probabilistic super-resolution of high-dimensional spatial fields using diffusion models is often computationally prohibitive due to the cost of operating directly in pixel space. We propose PODiff, a structured conditional generative framework that performs diffusion in a fixed, variance-ordered Proper Orthogonal Decomposition (POD) coefficient space, exploiting the orthogonality of POD modes to impose an interpretable, variance-ordered latent geometry. This design enables efficient ensemble generation, preserves dominant spatial structure, and yields spatially interpretable, well-calibrated uncertainty at substantially lower computational cost. We evaluate PODiff on sea surface temperature downscaling over the West Australian coast and on a controlled advection–diffusion benchmark. PODiff achieves reconstruction accuracy comparable to pixel-space and learned-latent diffusion while using a substantially smaller structured latent space, requiring significantly less memory, and producing more reliable uncertainty estimates than deterministic and Monte Carlo Dropout baselines.

## 1. Introduction

High-resolution spatial fields play a central role in scientific applications such as climate modeling, oceanography, geophysical flows, and numerical solutions of partial differential equations, but resolving fine-scale structure remains computationally demanding. Super-resolution methods aim to recover fine-scale structure from low-resolution inputs, but

[1]School of Earth and Oceans, UWA Oceans Institute, University of Western Australia, Crawley, WA, Australia [2]School of Physics, Mathematics and Computing, Computer Science and Software Engineering, University of Western Australia, Crawley, WA, Australia. Correspondence to: Onkar Jadhav <onkar.jadhav@uwa.edu.au>.

*Proceedings of the 43rd International Conference on Machine Learning*, Seoul, South Korea. PMLR 306, 2026. Copyright 2026 by the author(s).

reliable uncertainty quantification is equally important for scientific analysis (Gneiting & Raftery, 2007), particularly in regimes with sharp gradients and localized extremes.

Diffusion-based generative models have recently emerged as a powerful framework for probabilistic super-resolution and conditional generation (Song & Ermon, 2019; Ho et al., 2020; Song et al., 2020a;b; Dhariwal & Nichol, 2021; Kingma et al., 2021; Nichol & Dhariwal, 2021). Their iterative denoising formulation enables the generation of diverse high-fidelity samples conditioned on low-resolution inputs. However, applying diffusion models directly in the pixel space becomes computationally prohibitive at the resolutions common in scientific domains, requiring large networks, substantial memory, and long sampling times for ensemble generation. For instance, recent work has begun to explore diffusion-based models for geophysical downscaling, demonstrating improved probabilistic performance over deterministic baselines (Price et al., 2023; Leinonen et al., 2023; Watt & Mansfield, 2024; Li et al., 2024a; Du et al., 2024; Haitsiukevich et al., 2024; Li et al., 2024b), but these approaches remain computationally demanding at high spatial resolution.

To alleviate the cost of pixel-space diffusion, latent diffusion approaches perform diffusion in a learned low-dimensional space obtained from autoencoders (Vahdat et al., 2021; Rombach et al., 2022; Leinonen et al., 2023). While effective for natural images, such nonlinear latent spaces lack a clear connection between latent noise and spatial variability, limiting interpretability and principled uncertainty propagation in scientific applications.

In contrast to learned nonlinear latent spaces, many scientific fields exhibit strong low-rank linear structure, motivating the use of reduced-order representations such as Proper Orthogonal Decomposition (POD). POD provides a variance-ordered orthonormal basis that compactly represents dominant spatial patterns and has been widely used for compression and reconstruction (Sirovich, 1987; Berkooz et al., 1993; Benner et al., 2015). Despite its success, the potential of POD as a structured latent space for diffusion-based probabilistic modeling has received limited attention. Indeed, recent work has begun integrating reduced-order representations with deep learning (Champion et al., 2019;

Lee & Carlberg, 2020; Pichi et al., 2024; Coscia et al., 2024). However, the use of POD as a latent space for diffusion models with analytic uncertainty propagation to physical space remains largely unexplored.

In this work, we propose *PODiff*, a probabilistic super-resolution framework that performs conditional diffusion in a variance-ordered POD coefficient space. By leveraging the orthogonality and linear reconstruction properties of POD, PODiff enables efficient ensemble generation and analytically propagates predictive uncertainty to physical space, yielding spatially structured and interpretable uncertainty estimates. More broadly, we show that diffusion models can be re-parameterized into variance-ordered linear subspaces, enabling efficient uncertainty modeling, mode-level inspection, and exact second-order uncertainty propagation without reliance on learned encoders. We demonstrate the effectiveness of PODiff for sea surface temperature downscaling over the Western Australian (WA) coast as the primary real-world application, and on a controlled advection–diffusion benchmark as a diagnostic sanity check for uncertainty behavior, achieving competitive reconstruction accuracy and improved uncertainty calibration at substantially lower computational cost than pixel-space diffusion.

**Contributions** This work makes the following contributions: (i) We introduce PODiff, a conditional diffusion framework operating in a fixed, variance-ordered POD coefficient space, enabling efficient super-resolution of scientific fields. (ii) We show that diffusion in POD space admits analytic uncertainty propagation to the physical domain, yielding spatially resolved uncertainty estimates without auxiliary uncertainty networks. (iii) Through large-scale SST downscaling and a controlled advection–diffusion benchmark, we demonstrate that PODiff achieves improved calibration and competitive reconstruction accuracy at substantially lower computational cost than pixel-space diffusion models. (iv) We report low-resolution consistency checks and release code with evaluation scripts to support reproducibility.

**Conflict of Interest Disclosure.** The authors declare no financial conflicts of interest related to this work.

# 2. Methodology

PODiff performs diffusion-based generative modeling in a reduced-order space defined by proper orthogonal decomposition, reducing the effective dimensionality from $d$ spatial degrees of freedom to $K$ coefficients while preserving dominant spatial variability (Figure 1). This approach enables efficient ensemble generation and provides geometrically structured uncertainty estimates through a variance-ordered latent representation.

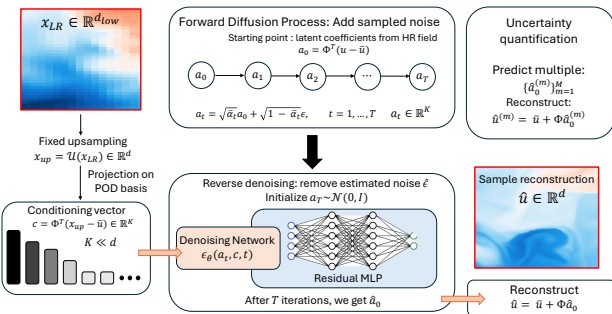

*Figure 1*. PODiff: conditional diffusion in a POD latent space. Low-resolution inputs are upsampled and projected onto a truncated POD basis to condition a diffusion model operating on POD coefficients. Reverse diffusion samples are reconstructed via the POD basis, yielding ensembles of high-resolution fields for uncertainty quantification.

## 2.1. Proper Orthogonal Decomposition as Latent Space

Let $\{u_i\}_{i=1}^N$ denote a collection of high-resolution spatial field snapshots, where $u_i \in \mathbb{R}^d$ represents a discretized field on a fixed grid with $d$ spatial points. We center the data by subtracting the empirical mean $\bar{u} = \frac{1}{N}\sum_{i=1}^N u_i$ and compute the POD basis by singular value decomposition of the centered snapshot matrix $U = [u_1 - \bar{u}, \ldots, u_N - \bar{u}]$.

Let $\{\lambda_k\}_{k=1}^d$ denote the eigenvalues of the empirical covariance matrix (i.e., the squared singular values from the POD/SVD), ordered such that $\lambda_1 \geq \lambda_2 \geq \cdots \geq \lambda_d$. The truncation level $K$ is selected as the smallest integer that satisfies

$$\frac{\sum_{k=1}^K \lambda_k}{\sum_{k=1}^d \lambda_k} \geq \eta,$$

where $\eta \in (0, 1)$ denotes a prescribed cumulative variance threshold.

The resulting POD basis $\Phi = [\phi_1, \ldots, \phi_K] \in \mathbb{R}^{d \times K}$ consists of $K$ orthonormal spatial modes ordered by explained variance, satisfying

$$\Phi^\top \Phi = I_K.$$

Any field $u$ can be approximated as

$$u \approx \bar{u} + \Phi a, \qquad a = \Phi^\top(u - \bar{u}), \qquad (1)$$

where $a \in \mathbb{R}^K$ denotes the corresponding POD coefficients.

POD provides an optimal rank-$K$ approximation by minimizing the mean squared reconstruction error (Sirovich, 1987; Berkooz et al., 1993). This hierarchical organization of variance yields an orthogonal, variance-ordered latent representation in which individual modes correspond to progressively finer-scale spatial structures. In the context of

diffusion modeling, this structure enables mode-level inspection and interpretable uncertainty analysis. Although POD coefficients are standardized during training for numerical stability, the underlying variance ordering of the POD basis remains central to the organization and interpretation of the learned latent dynamics (Brunton & Kutz, 2022).

## 2.2. Conditional Diffusion in POD Space

**Conditioning mechanism.** Let $x_{\mathrm{LR}} \in \mathbb{R}^{d_{\mathrm{low}}}$ denote a low-resolution observation of the same physical field defined on a coarse spatial grid, where $d_{\mathrm{low}} \ll d$. First, the low-resolution input is mapped to the high-resolution grid using a fixed bicubic upsampling operator

$$x_{\mathrm{up}} = \mathcal{U}(x_{\mathrm{LR}}) \in \mathbb{R}^d.$$

Second, the upsampled field is projected onto the retained POD subspace via

$$c = \Phi^\top (x_{\mathrm{up}} - \bar{u}) \in \mathbb{R}^K.$$

This projection yields a consistent $K$-dimensional conditioning vector, even when the upsampled input does not lie entirely within the span of the retained POD modes.

**Forward diffusion process.** Let $a_0 \in \mathbb{R}^K$ denote the standardized POD coefficients of a target high-resolution field. The forward diffusion process progressively adds Gaussian noise (Ho et al., 2020; Song et al., 2020b) over $T$ discrete timesteps to $a_0$ as

$$q(a_t \mid a_0) = \mathcal{N}\big(a_t; \sqrt{\bar{\alpha}_t}\, a_0, (1 - \bar{\alpha}_t)I\big),$$

where $\{\alpha_t\}_{t=1}^T$ defines a variance schedule and $\bar{\alpha}_t = \prod_{s=1}^t \alpha_s$. This formulation admits a closed-form sampling expression

$$a_t = \sqrt{\bar{\alpha}_t}\, a_0 + \sqrt{1 - \bar{\alpha}_t}\, \epsilon, \qquad \epsilon \sim \mathcal{N}(0, I).$$

**Reverse diffusion process.** A neural network $\epsilon_\theta(a_t, c, t)$ is trained to predict the injected noise given the noisy coefficients $a_t$, the conditioning vector $c$, and the timestep $t$. The training objective is given by

$$\mathcal{L}(\theta) = \mathbb{E}_{a_0, t, \epsilon} \left[ \|\epsilon - \epsilon_\theta(a_t, c, t)\|_2^2 \right],$$

where $t$ is sampled uniformly from $\{1, \ldots, T\}$ and $\epsilon \sim \mathcal{N}(0, I)$.

At inference time, given a conditioning vector $c$ obtained from a low-resolution input, samples are drawn from the learned conditional distribution $p_\theta(a \mid c)$ by initializing $a_T \sim \mathcal{N}(0, I)$ and iteratively applying the reverse diffusion process to obtain samples $\hat{a}_0$.

The predicted coefficients $\hat{a}_0$ are inverse-standardized and mapped back to physical space via

$$\hat{u} = \bar{u} + \Phi \hat{a}_0.$$

Repeating this procedure with independent noise realizations yields an ensemble of reconstructions, enabling Monte Carlo estimation of predictive uncertainty.

## 2.3. Denoising Network Architecture

The denoising network $\epsilon_\theta$ is implemented as a residual multilayer perceptron operating on $K$-dimensional latent representations. The network takes as input the concatenation of the noisy coefficients $a_t$ and the conditioning vector $c$, together with an embedding of the diffusion timestep $t$, which is added to each hidden layer. The architecture consists of multiple residual blocks with shared hidden dimensionality, followed by a linear projection back to $\mathbb{R}^K$ to predict the noise vector.

## 2.4. Baselines and Ablations

We compare PODiff with multiple baselines that isolate different components of the approach. Implementation and training details for all baselines are provided in Section 3.

**POD Projection (deterministic latent baseline).** We consider a deterministic baseline obtained by directly reconstructing from the projected conditioning signal without learning or stochastic modeling. Specifically, given the conditioning coefficients $c$, the reconstructed field is

$$\hat{u}_{\mathrm{proj}} = \bar{u} + \Phi c = \bar{u} + \Phi \Phi^\top (x_{\mathrm{up}} - \bar{u}). \qquad (2)$$

This baseline employs the same POD basis and upsampling operator as PODiff but does not involve diffusion, parameter learning, or uncertainty estimation. Therefore, it isolates the effect of variance-based dimensionality reduction alone.

**RandOrthDiff (latent basis ablation).** To assess the role of the POD basis, we introduce an ablation in which the POD modes are replaced by a randomly sampled orthonormal basis $\Psi \in \mathbb{R}^{d \times K}$. The diffusion architecture, conditioning mechanism, noise schedule, and training procedure are kept identical to PODiff. This ablation isolates the effect of the latent basis by keeping the diffusion architecture and training procedure fixed while replacing POD modes with a random orthonormal basis.

**Deterministic U-Net.** We include a convolutional U-Net (Ronneberger et al., 2015) trained with a mean squared error loss to map low-resolution inputs directly to high resolution outputs. This baseline represents a standard deterministic learning-based approach for super-resolution.

**MC Dropout U-Net.** As a probabilistic baseline, MC Dropout applies dropout (rate 0.2) to encoder and decoder convolutional blocks during training and inference, with uncertainty estimated from an ensemble of stochastic forward passes (Gal & Ghahramani, 2016).

**Pixel-space diffusion (PixelDiff).** PixelDiff uses the same convolutional U-Net backbone as the deterministic U-Net baseline but is trained with a denoising diffusion objective. The diffusion timestep is incorporated via standard sinusoidal time embeddings added to each residual block, and conditioning on the low-resolution input is performed by concatenating the bicubically upsampled input with the noisy high-resolution field along the channel dimension. Stochastic predictions are generated via iterative denoising, enabling ensemble-based uncertainty estimation in pixel space (Ho et al., 2020; Dhariwal & Nichol, 2021).

**VAE-based latent diffusion (VAE-LDM).** To compare PODiff against learned latent representations, we include a VAE-based latent diffusion baseline following standard latent diffusion practice. A convolutional encoder–decoder compresses each high-resolution field into an $8 \times 8 \times 40$ spatial latent tensor, corresponding to 2560 latent features. A conditional U-Net diffusion model is then trained in this learned latent space using the same training data and diffusion schedule as the pixel-space diffusion baseline. This baseline tests whether a learned nonlinear latent space can match the accuracy and calibration of PODiff while reducing the cost of pixel-space diffusion.

**RBF interpolation baseline.** We include radial basis function (RBF) interpolation as a classical, non-learning baseline for SST downscaling. RBF interpolation directly maps the low-resolution observation to the high-resolution grid using a thin-plate spline kernel, without training or data-driven parameter learning. Bicubic interpolation is used only to align low-resolution inputs to the high-resolution grid for learning-based models and loss computation. The RBF baseline is reported separately as a classical interpolation method.

## 2.5. Uncertainty Quantification

The predictive uncertainty in PODiff is estimated from ensembles of samples generated by the latent diffusion model. At inference time, multiple realizations of the POD coefficients $\{\hat{a}_0^{(m)}\}_{m=1}^M$ are obtained by independent reverse diffusion trajectories conditioned on the same low-resolution input. Each sample is mapped to the physical space through linear reconstruction $\hat{u}^{(m)} = \bar{u} + \Phi \hat{a}_0^{(m)}$.

Due to the linearity of the reconstruction operator, the spatial covariance of the predictive distribution admits the closed-form expression

$$\Sigma_u = \Phi \Sigma_a \Phi^\top, \qquad (3)$$

where $\Sigma_a$ denotes the empirical covariance of the latent coefficient ensemble. This structure reflects that uncertainty in leading POD coefficients primarily affects large-scale spatial patterns, while uncertainty in higher-order coefficients

tends to manifest as more localized or fine-scale variability.

We assess the quality of the resulting predictive uncertainty using multiple complementary metrics. Empirical coverage evaluates the fraction of ground-truth values contained within nominal predictive intervals. Reliability curves compare nominal and empirical coverage levels across a range of confidence thresholds. The calibration error is summarized using the mean absolute calibration error (MACE), defined as the average absolute deviation between nominal and empirical coverage. In addition, we report the continuous ranked probability score (CRPS), which provides a proper scoring rule that jointly evaluates sharpness and calibration of the predictive distribution (Gneiting & Raftery, 2007).

## 2.6. Computational Advantage

PODiff achieves efficiency by operating entirely in a reduced-order latent space, where diffusion is performed on $K \ll d$ coefficients rather than full-resolution fields. This substantially reduces parameter count, memory footprint, and sampling cost relative to pixel-space generative models. Moreover, because reconstruction is linear, ensemble statistics propagate analytically from latent to spatial space, enabling efficient uncertainty estimation without repeated full-network evaluations.

**Design rationale: POD vs. learned latents.** We use POD rather than learned autoencoders for several reasons. First, POD does not require encoder–decoder training, avoiding additional optimization complexity and latent-space distortions introduced by nonlinear decoders. Second, the POD basis is orthonormal and variance-ordered by construction, yielding a stable latent geometry for diffusion modeling. Third, the ordered modes support mode-level inspection of spatial scales. Finally, linear reconstruction provides a direct relationship between latent and spatial second-order statistics via $\Sigma_u = \Phi \Sigma_a \Phi^\top$, which is less direct when reconstruction is nonlinear. For spatially coherent fields with dominant low-rank structure, common in geophysical flows, climate data, and many PDE solutions, POD offers a stable and efficient latent representation without requiring end-to-end training of additional generative components.

Accordingly, while learned latent diffusion models are a powerful alternative, we do not pursue autoencoder-based latent spaces here, as our focus is on uncertainty interpretability and analytic propagation, along with competitive reconstruction quality, rather than representation learning.

## 2.7. Limitations

PODiff is most effective when target fields admit low-rank linear structure. Performance may degrade for highly turbulent or discontinuous fields requiring many modes, for

systems with strong nonlinear interactions, or under significant distributional shift. Because the POD basis is fixed after decomposition, adapting to such changes may require recomputing the basis or incorporating adaptive reduced-order representations, which we leave to future work. Also, truncation uncertainty is not modeled; however, retained modes capture $\geq$99% variance. This work also focuses on single-variable scalar fields. PODiff can be extended to multivariate fields by constructing a joint POD basis over normalized concatenated variables, but variables with different units, scales, and cross-variable correlations require careful normalization and validation. We therefore leave multivariate PODiff to future work.

## 3. Experimental Setup

We evaluate PODiff on a real-world sea surface temperature downscaling task over the coast of Western Australia and on a controlled advection-diffusion problem.

### 3.1. SST Downscaling

For the SST task, we selected the calendar year 2011 as a dedicated test period because it contains a well-documented marine heatwave event along the West Australian coast, characterized by elevated sea surface temperatures and sharp spatial gradients. This makes 2011 a scientifically meaningful stress test for both reconstruction accuracy and uncertainty calibration, beyond average climatological conditions. We train all models using data from 1998-2009, validate hyperparameters and model selection on the year 2010, and report all quantitative results on the held-out test year 2011. This corresponds to approximately 4000 daily training samples, 365 validation samples from 2010, and 365 test samples from 2011.

**High resolution data.** We used daily SST fields on a fixed WA coastal window with a target resolution of $640 \times 480$ obtained from a Regional Ocean Modeling System (ROMS) (Shchepetkin & McWilliams, 2005). Land points are masked, and all metrics are computed over ocean pixels only. The POD basis is computed exclusively from training high resolution data.

**Low-resolution inputs.** Low-resolution inputs are obtained from the ACCESS-S2 ocean reanalysis model (Wedd et al., 2022), which provides SST fields at a native spatial resolution of $53 \times 31$ over the same WA coastal domain used to extract high-resolution SST fields from the ROMS model. This corresponds to an approximately $12\times$ spatial upscaling in each direction and a pixel-count ratio of approximately $187\times$ between the low- and high-resolution grids. To enable direct comparison and pixelwise loss evaluation, the ACCESS-S2 fields are interpolated onto the ROMS

$640 \times 480$ grid using bicubic interpolation. This interpolation step is used solely for grid alignment and does not introduce additional fine-scale information.

**POD representation.** High-resolution SST fields are projected onto the POD basis described in Section 2.1. The latent dimension $K$ is treated as a hyperparameter and varied in our experiments to study the effect of latent truncation. We study $K \in \{10, 20, 40\}$ to test truncation sensitivity and confirm stability. Unless stated otherwise, results use $K = 40$, which captures approximately 99% of the cumulative variance in the training data.

**PODiff.** PODiff models the conditional distribution $p(a \mid c)$ using latent diffusion in coefficient space, where the conditioning vector $c$ is obtained by projecting the upsampled low-resolution field onto the retained POD basis. The denoiser is a compact conditional MLP. In all experiments, the MLP uses four hidden layers with width 256 and sinusoidal timestep embeddings. We train the diffusion model with $T = 1000$ diffusion steps and generate samples at inference using a reduced-step sampler with $S = 100$ denoising steps. Unless otherwise stated, uncertainty estimates use $M = 100$ samples.

**Coefficient normalization.** Both target POD coefficients $a$ and conditioning coefficients $c$ are standardized per mode using training-set statistics prior to diffusion training. Sampling and reconstruction are performed by de-normalizing the generated coefficients. This standardization is a practical training choice and preserves the variance-ordered POD structure.

**Training.** All models are trained using AdamW with learning rate $2 \times 10^{-4}$. Diffusion models are selected based on validation diffusion loss, while deterministic baselines are selected based on validation RMSE, reflecting their respective training objectives.

**Pixel-space U-Net.** As a deterministic learning-based baseline, we instantiate the U-Net described in Section 2.4 as a standard 2D architecture operating directly on the $640 \times 480$ grid. The network follows a symmetric encoder-decoder architecture with four resolution levels, a base channel width of $C = 128$ (channels per level $C, 2C, 4C, 8C$), and skip connections between corresponding encoder and decoder stages. The model is trained using an $\ell_2$ regression loss to predict high-resolution SST fields from interpolated low-resolution inputs. Additionally, to provide a learning-based uncertainty baseline, we apply Monte Carlo Dropout to the pixel-space U-Net by enabling dropout layers at inference time and generating an ensemble of stochastic forward passes. Unless otherwise stated, MC Dropout uncertainty estimates also use $M = 100$ samples.

To test whether the performance of the pixel-space U-Net is sensitive to model capacity, we additionally evaluate a reduced-capacity U-Net with identical depth and skip-connection structure but a smaller base width $C = 32$.

**PixelDiff.** Unless otherwise stated, PixelDiff is trained with $T = 1000$ diffusion steps and uses $S = 100$ denoising steps at inference, with uncertainty estimates obtained from $M = 100$ samples.

**VAE-LDM.** The learned-latent diffusion baseline uses a convolutional encoder–decoder with base width $C = 64$ and an $8 \times 8 \times 40$ latent tensor. The latent diffusion model uses a U-Net denoiser with base width $C = 128$, matching the PixelDiff backbone.

**Metrics.** We report RMSE and MAE for reconstruction accuracy over the full test set and over extreme SST events. Extreme events are defined as exceedances of the 90th percentile of the day-of-year climatological SST. Uncertainty quality is assessed using empirical coverage, reliability curves, and mean absolute calibration error (MACE) over nominal levels $\{50\%, 70\%, 90\%, 95\%\}$.

**Uncertainty evaluation protocol.** Uncertainty metrics are computed by averaging ensemble-based statistics over 20 test days selected from the January-March 2011 marine heatwave period to limit computational cost, while reconstruction metrics are evaluated over the full 2011 test year. We verified that increasing the ensemble size beyond $M = 100$ does not significantly change coverage estimates (Appendix B, Table 4).

### 3.2. Advection–Diffusion Problem

We additionally evaluate PODiff on a controlled two-dimensional advection–diffusion problem (Raissi et al., 2019; Evans, 2010) to analyze reconstruction accuracy and uncertainty behavior in a setting with known governing dynamics. We solve the two-dimensional linear advection–diffusion equation

$$\partial_t u + v_x \partial_x u + v_y \partial_y u = \kappa (\partial_{xx} u + \partial_{yy} u),$$

on a periodic square domain. The velocity components are sampled as $v_x, v_y \sim \mathcal{U}[-1, 1]$, and the diffusivity is sampled log-uniformly as $\kappa \sim \log \mathcal{U}(10^{-4}, 5 \times 10^{-3})$. Synthetic data are generated by numerically integrating the linear advection–diffusion equation with periodic boundary conditions from randomized smooth initial conditions, using advection velocities sampled uniformly from $[-1, 1]$ in each direction and diffusivity coefficients sampled log-uniformly from $[10^{-4}, 5 \times 10^{-3}]$. The dataset consists of 500 simulated trajectories, each recorded at four snapshot times (steps 50, 100, 150, and 200). High-resolution fields

are defined on a $128 \times 128$ grid. Low-resolution inputs are obtained by block-averaging to a $32 \times 32$ grid and then upsampling to the high-resolution grid. Data are split at the trajectory level, with 20% held out for testing. For uncertainty evaluation, we randomly select 20 test snapshots and generate ensembles of $M = 100$ samples. For this benchmark, PODiff uses a latent dimension $K = 40$.

## 4. Results

We first present results for SST downscaling over the Western Australian coast, followed by a controlled advection–diffusion problem to examine uncertainty behavior.

For the SST experiments, PODiff is compared with interpolation-based and learning-based baselines using both reconstruction accuracy and uncertainty quantification metrics. Reconstruction accuracy, measured using RMSE and MAE, is reported in Table 1, while representative spatial error patterns are shown in Figure 2. The quality of the uncertainty estimates is evaluated using empirical coverage statistics reported in Table 2, reliability curves shown in Figure 3, and spatial maps of predictive uncertainty illustrated in Figure 4. Results for the advection–diffusion experiment are presented in Section 4.3. All learning-based methods are trained and evaluated on identical datasets. Additional ablations and supplementary results are provided in Appendices A, B, and C.

### 4.1. SST Downscaling: Reconstruction Accuracy

We first evaluate the reconstruction accuracy of the downscaled SST using RMSE and MAE, with quantitative results summarized in Table 1. The test set includes all days in 2011, covering the full annual cycle and associated extreme events. PODiff achieves the lowest error across all reported metrics, both when evaluated over the full test set and when restricted to extreme SST events. For example, PODiff-K40 achieves a global RMSE of 0.3923 °C and a MAE of 0.2976 °C, compared to 0.6788 °C / 0.5141 °C for U-Net and 0.7783 °C / 0.5804 °C for RBF interpolation. Using a random orthonormal basis in place of POD modes substantially degrades performance, with RandOrthDiff yielding an RMSE close to 1.0 °C, despite sharing the same diffusion architecture and latent dimensionality. Moreover, a reduced-capacity U-Net achieves errors comparable to the larger U-Net, indicating that the performance gap is not driven by model capacity but by limitations of deterministic pixel-space learning. Errors increase for all methods during extreme events, but the relative ranking remains unchanged. PODiff-K40 achieves an extreme-event RMSE of 0.4836 °C, compared to 0.7899 °C for RBF interpolation and 0.8366 °C for the U-Net baseline. RandOrthDiff exhibits the largest degradation, with an RMSE of 1.2309 °C under extreme conditions. Despite sharing the same diffusion architec-

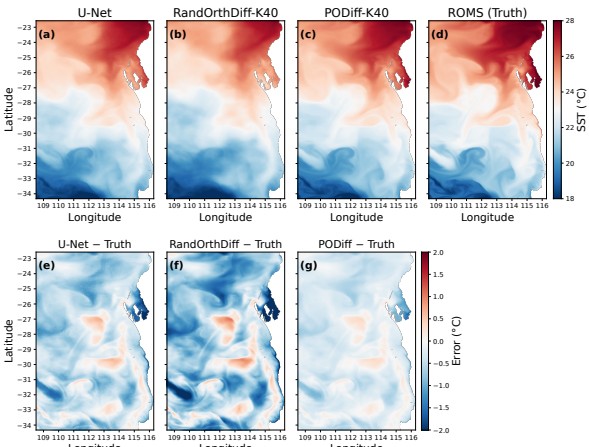

*Figure 2.* Qualitative comparison of SST downscaling methods for a representative test day (31 January 2011, randomly selected for visualization). Top row (a-d): reconstructed SST fields from U-Net, RandOrthDiff-K40, PODiff-K40, and ROMS ground truth. Bottom row (e-g): corresponding signed reconstruction errors (prediction minus truth). PODiff-K40 achieves the lowest reconstruction errors, particularly in regions of strong thermal gradients and close to the coast. The RandOrthDiff employs the same diffusion architecture and latent dimensionality as PODiff but replaces POD modes with a random orthonormal basis for a controlled comparison of latent representations.

ture and latent dimensionality as PODiff, replacing POD modes with a random orthonormal basis leads to substantially degraded and less stable reconstructions, highlighting the importance of a data-adaptive latent representation.

We additionally evaluate a deterministic POD-projection (POD-proj) baseline that reconstructs fields directly from projected conditioning coefficients, without diffusion or stochastic modeling. Although it operates in the same reduced latent space as PODiff, this baseline yields higher errors (RMSE 0.7084 °C, MAE 0.5223 °C), indicating that dimensionality reduction alone is insufficient for accurate reconstruction.

For completeness, we also evaluate a pixel-space diffusion (PixelDiff) model operating directly on the $640 \times 480$ grid. PixelDiff achieves reconstruction accuracy comparable to PODiff, with an RMSE of 0.4118 °C and an MAE of 0.3158 °C, and competitive performance under extreme events. However, this accuracy is obtained at substantially higher computational cost, as quantified in Table 3.

We also compare against a learned-latent diffusion baseline (VAE-LDM). VAE-LDM obtains an RMSE of 0.4011, CRPS of 0.2915, and 90% coverage of 0.9022, which is close to PODiff-K40. However, VAE-LDM uses a 2560-dimensional learned latent representation, compared with 40 POD coefficients for PODiff, and requires approximately 39M parameters and 28 hours of two-stage training. In contrast, PODiff uses 0.20M parameters and trains in 3.8 hours.

*Table 1.* Reconstruction error metrics for SST downscaling. Lower values indicate better performance. The test set consists of all days in the calendar year 2011, ensuring the evaluation of the entire annual cycle, including seasonal transitions and extreme events.

| MODEL | RMSE ↓ | MAE ↓ | EXTREME RMSE ↓ | EXTREME MAE ↓ |
|---|---|---|---|---|
| PODIFF-K40 | 0.3923 | 0.2976 | 0.4836 | 0.3537 |
| PODIFF-K20 | 0.5171 | 0.3923 | 0.6373 | 0.4661 |
| PODIFF-K10 | 0.7725 | 0.5861 | 0.9521 | 0.6964 |
| POD-PROJ | 0.7084 | 0.5223 | 0.8896 | 0.6305 |
| PIXELDIFF | 0.4118 | 0.3158 | 0.4899 | 0.3600 |
| VAE-LDM | 0.4011 | 0.3005 | 0.4889 | 0.3591 |
| U-NET | 0.6788 | 0.5141 | 0.8366 | 0.6109 |
| U-NET (REDUCED) | 0.6819 | 0.5273 | 0.8415 | 0.6111 |
| RBF | 0.7784 | 0.5804 | 0.7899 | 0.5936 |
| RANDORTHDIFF | 0.9987 | 0.7577 | 1.2309 | 0.9003 |

*Note:* All metrics averaged over 365 test days. Standard deviations across 5 training runs are <0.01 for all models.

These results indicate that PODiff's advantage is not merely due to leaving pixel space, but also due to the compact and structured POD representation.

Figure 2 shows a qualitative comparison of reconstructed SST fields and corresponding error maps for a representative day (31 January 2011). Panels (a–d) display reconstructed SST fields from U-Net, RandOrthDiff-K40, PODiff-K40, and the ROMS ground truth, respectively, while panels (e–g) show signed reconstruction errors relative to ROMS. The U-Net reduces the overall error magnitude with deviations mostly within ±0.6°C, but shows a widespread low-amplitude bias, consistent with oversmoothing. RandOrthDiff produces larger, spatially coherent residuals, consistent with its higher RMSE. In contrast, PODiff (panel (g)) exhibits the smallest error magnitude, with errors that are spatially sparse and largely confined to regions of strong thermal gradients.

Together, Table 1 and Figure 2 indicate that the advantage of PODiff arises not only from a lower aggregate error but also from a qualitatively more structured and spatially localized distribution of reconstruction errors.

Beyond reconstruction accuracy, PODiff provides limited interpretability through its reduced-order representation. Leading POD modes capture large-scale SST patterns, while higher-order modes represent finer spatial variability. The first POD mode alone explains over 70% of the total variance as shown in Appendix A, Fig. 6.

### 4.2. SST Downscaling: Uncertainty Quantification

We next evaluate the quality of the uncertainty estimates produced by PODiff using empirical coverage, reliability curves, and spatially resolved predictive variance. Quantitative coverage statistics are summarized in Table 2, while Figures 3 and 4 illustrate reliability curves and spatial calibration behavior.

As shown in Table 2, PODiff-K40 achieves close agreement

*Table 2.* Empirical coverage at different nominal confidence levels. Values are reported as coverage (absolute deviation from nominal).

| Nominal Level | PODiff-K40 | MC Dropout U-Net | PixelDiff |
|---|---|---|---|
| 50% | 0.4717 (0.0283) | 0.4111 (0.0889) | 0.4658 (0.0342) |
| 70% | 0.6849 (0.0151) | 0.6508 (0.0492) | 0.6799 (0.0201) |
| 90% | 0.9009 (0.0009) | 0.8871 (0.0129) | 0.9010 (0.0010) |
| 95% | 0.9571 (0.0071) | 0.9401 (0.0099) | 0.9551 (0.0051) |

with nominal coverage, particularly at higher confidence levels. PixelDiff exhibits similarly well-calibrated behavior across all levels, while the MC Dropout U-Net shows systematic undercoverage, with the largest deviations occurring at lower confidence intervals (Gal & Ghahramani, 2016). For PODiff, the 90% and 95% intervals closely match their targets, with empirical coverages of 0.9009 and 0.9571, respectively. Mild undercoverage is observed at lower nominal levels (e.g., 0.4717 at 50%), indicating slightly overconfident central intervals. This behavior may plausibly arise from truncation of higher-order POD modes or conservative diffusion noise schedules that prioritize tail calibration. Despite this effect, the overall calibration error remains low, with a mean absolute calibration error of 0.0128 averaged across all levels. Consistent with these findings, PODiff-K40 achieves substantially improved probabilistic accuracy relative to MC Dropout U-Net, as reflected by a lower CRPS (0.2889 vs. 0.4821) on the same 20 test days. This pattern

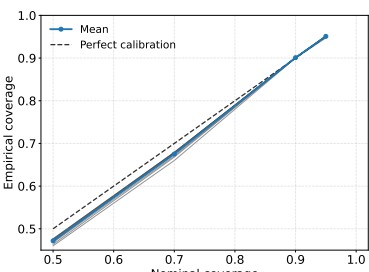

*Figure 3.* Reliability curves for PODiff showing empirical coverage as a function of nominal confidence level, computed using ensembles of 100 samples per day and averaged over 20 randomly selected test days. The thick curve denotes the mean reliability across days, while thin curves correspond to individual test days.

is also visible in the reliability curves in Figure 3. At lower nominal levels, the mean curve lies below the diagonal reference, indicating that the predicted intervals are slightly too narrow. As the confidence level increases, the curve approaches the ideal reference and is nearly aligned at 90% and above. The reliability curves for individual test days cluster tightly around the mean, suggesting that this behavior is consistent over time rather than driven by a small number of atypical cases. The spatial structure of the predictive uncertainty is illustrated in Figure 4 through the posterior standard deviation. The higher uncertainty is concentrated near coastal regions and areas with strong temperature gra-

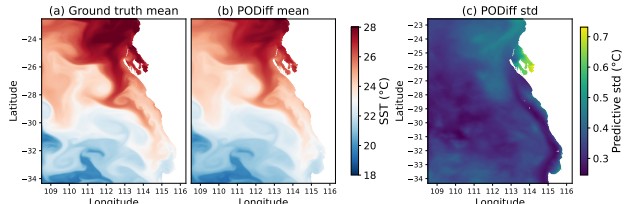

*Figure 4.* Spatial distribution of predictive uncertainty for PODiff, shown as the posterior standard deviation of ensemble predictions averaged over 20 randomly selected test days, with 100 samples generated per day.

| Method | Params | Peak GPU Mem | Training Time | Inference Time per Sample |
|---|---|---|---|---|
| U-Net | 33M | 8.8 GB | 8.2 h | 0.05 s |
| PODiff (K=40) | 0.20M | 1.4 GB | 3.8 h | 0.08 s |
| RandOrthDiff (K=40) | 0.20M | 1.4 GB | 3.8 h | 0.08 s |
| PixelDiff | 33M | 12.5 GB | 48 h | 1.24 s |
| VAE-LDM | 39M | 13.1 GB | 28 h | 0.88 s |

*Table 3.* Comparison of computational cost for SST downscaling at full resolution ($640 \times 480$). PODiff and RandOrthDiff operate in a reduced POD latent space ($K = 40$), resulting in substantially lower parameter count and peak GPU memory compared to a pixel-space deterministic U-Net and diffusion model (PixelDiff). Inference time for diffusion-based methods is reported per generated sample (including all denoising steps), whereas U-Net inference corresponds to a single deterministic forward pass. All experiments were conducted on the Setonix supercomputer using AMD Instinct MI250X GPUs.

dients, while open-ocean regions exhibit lower variance. In particular, these uncertainty patterns do not strictly mirror reconstruction errors, but instead highlight regions where small-scale variability and unresolved dynamics are most prominent. This suggests that PODiff assigns uncertainty in a spatially meaningful way rather than producing uniform or noise-dominated variance fields.

Together, these results demonstrate that PODiff yields well-calibrated uncertainty at practically relevant confidence levels with spatially structured uncertainty. Spatial calibration error maps at 50% and 90% are shown in Appendix B (Fig. 7), indicating mild and spatially coherent miscalibration.

Table 3 compares the computational cost of all methods at full spatial resolution ($640 \times 480$). The deterministic U-Net achieves the lowest inference time per forward pass, but produces only a single point estimate. In contrast, PODiff and RandOrthDiff employ diffusion-based sampling and therefore incur higher per-sample inference cost, which scales linearly with the number of generated ensemble members. Crucially, by operating in a low-dimensional POD latent space ($K = 40$), PODiff achieves diffusion-based uncertainty modeling with orders-of magnitude fewer parameters and substantially lower peak GPU memory than pixel-space diffusion. While PixelDiff attains comparable reconstruc-

tion accuracy, it requires approximately $13\times$ longer training time and an order-of-magnitude higher inference cost per sample. These results highlight that PODiff enables probabilistic super-resolution with a per-sample inference cost close to that of deterministic models, with total ensemble cost scaling linearly in the number of samples, while avoiding the prohibitive expense of pixel-space diffusion.

### 4.3. Advection–Diffusion Problem

We next report results on the advection–diffusion problem, focusing on reconstruction accuracy and the calibration of predictive uncertainty. This benchmark is used as a controlled diagnostic setting to analyze uncertainty behavior rather than as a full comparative evaluation across baselines. Figure 5 illustrates the ensemble mean and predictive uncer-

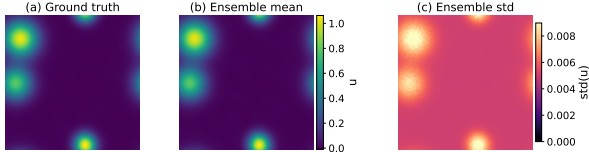

*Figure 5.* Advection–diffusion uncertainty example. (a) Ground-truth solution for a representative test snapshot of the two-dimensional advection–diffusion problem. (b) Ensemble mean obtained from $S = 100$ diffusion samples. (c) Posterior standard deviation across the ensemble.

tainty for a representative advection-diffusion test snapshot. The ensemble mean closely matches the ground-truth solution, accurately recovering both locations and amplitudes of the localized structures, which confirms that stochastic sampling does not introduce bias or oversmoothing.

The posterior standard deviation reveals a highly localized uncertainty pattern. Elevated uncertainty is concentrated around the sharp, localized peaks in the solution, while the surrounding regions exhibit uniformly low variance. This behavior indicates increased epistemic uncertainty near sharp, localized features, while the smooth background remains well constrained. Importantly, the uncertainty field is spatially selective and structured, rather than diffuse or noise-dominated, indicating that PODiff captures meaningful solution-dependent uncertainty in this controlled PDE setting. Reliability curves and empirical coverage statistics for this benchmark are reported in Appendix C.

In addition to the qualitative uncertainty structure shown in Figure 5, PODiff achieves accurate ensemble means on the advection–diffusion problem. Averaged over the evaluated test snapshots, the ensemble mean produces an RMSE of 0.018 and a MAE of 0.0098 relative to the ground truth, indicating that stochastic sampling does not degrade the reconstruction accuracy in this controlled PDE setting.

Overall, the advection–diffusion experiment complements the SST results by demonstrating that PODiff produces ac-

curate ensemble means and spatially localized, interpretable uncertainty in a controlled PDE setting.

### Impact Statement

This work advances uncertainty-aware super-resolution of spatial fields, with direct applications in climate and geophysical modeling. For example, probabilistic downscaling of ocean temperature or atmospheric variables can support more reliable regional forecasts, extreme-event analysis, and risk-aware decision-making in environmental monitoring and resource management. Reliable uncertainty estimates are essential for scientific interpretation and downstream use. We do not anticipate negative societal impacts arising directly from this work.

### Code Availability

Code for POD fitting, diffusion training, inference, and evaluation is available at: https://github.com/OnkaraJadhav/PODiff.git.

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

# A. POD Modes and Mode-Level Inspection

**Purpose.** This appendix provides qualitative context for the Proper Orthogonal Decomposition representation used in PODiff. It illustrates how variance and spatial structure are distributed across POD modes. Figure 6 illustrates the temporal

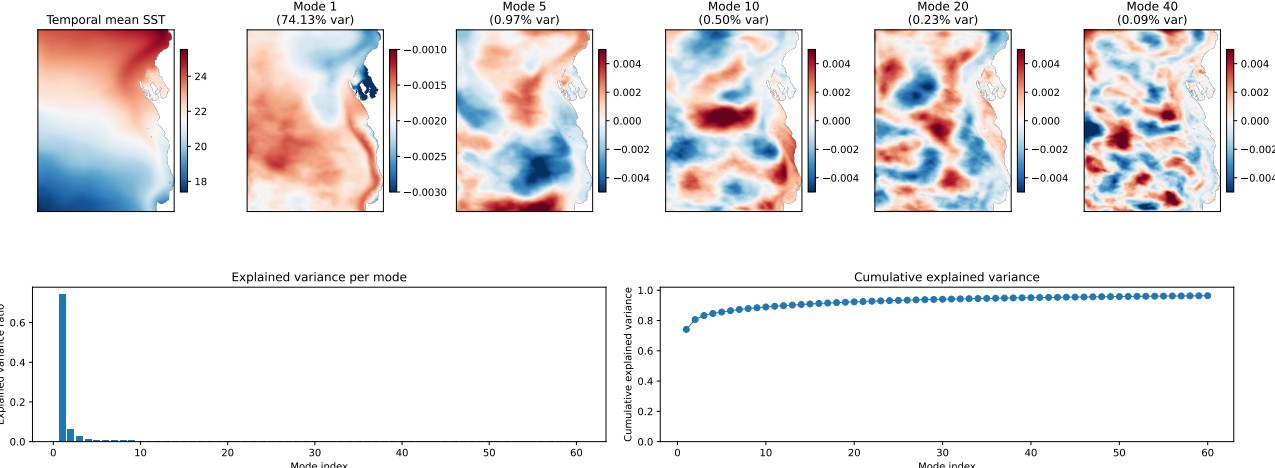

*Figure 6.* Temporal mean SST field (left), selected POD spatial modes (Modes 1, 5, 10, 20, and 40), and the associated explained-variance spectrum with cumulative variance. Lower-index modes capture dominant large-scale structure, while higher-index modes exhibit increasingly localized spatial variability.

mean SST field together with selected POD spatial modes (Modes 1, 5, 10, 20, and 40) and the associated variance spectrum. The leading POD mode captures the dominant basin-scale SST structure and explains approximately 74% of the total variance in the training data. Subsequent modes contribute progressively smaller fractions of variance and exhibit increasingly localized and oscillatory spatial patterns.

The explained-variance spectrum shows a rapid decay after the first few modes, followed by a long tail associated with fine-scale variability. As shown by the cumulative variance curve, approximately 99% of the total variance is retained by $K = 40$ modes, which motivates the truncation level used in the main experiments.

Beyond dimensionality reduction, the variance-ordered POD basis provides a limited and descriptive form of interpretability through scale separation. Lower-index modes predominantly represent large-scale, smooth spatial organization, while higher-index modes encode progressively finer-scale features and sharper gradients. This ordering offers qualitative insight into how large-scale and small-scale variability are distributed across the latent representation. We emphasize that this interpretation is purely descriptive and does not imply physical causality or direct correspondence between individual modes and underlying dynamical processes.

# B. Spatial Reliability Analysis and Ensemble Size Sensitivity

**Purpose.** This appendix examines the spatial structure of calibration error and the sensitivity of empirical coverage to ensemble size. It provides supplementary evidence that calibration errors are spatially coherent and that moderate ensemble sizes yield stable uncertainty estimates.

### B.1. Spatial calibration error maps

Figure 7 shows spatial maps of empirical coverage minus nominal coverage for PODiff-K40 at the 50% and 90% confidence levels, averaged over the same test days used for the uncertainty metrics in Table 2. At the 50% level, mild undercoverage is observed across much of the domain, consistent with the slight undercoverage reported in the reliability curves. Importantly, the deviations are smooth and spatially coherent, rather than dominated by localized or noisy artifacts.

At the 90% level, calibration error is close to zero across most of the domain, indicating well-calibrated high-confidence predictive intervals. These spatial patterns are consistent with the aggregate reliability behavior reported in Figure 3, and indicate that remaining miscalibration is modest and structured rather than random.

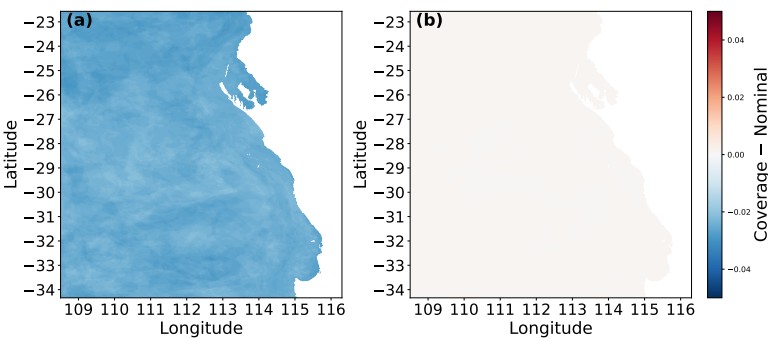

*Figure 7.* Spatial maps of empirical coverage minus nominal coverage for PODiff-K40 at the 50% and 90% confidence levels, averaged over the test set. Warm (cool) colors indicate overcoverage (undercoverage).

### B.2. Effect of ensemble size

Table 4 reports empirical coverage as a function of ensemble size $M$, averaged over 20 test days. Increasing the ensemble size from $M = 50$ to $M = 100$ leads to small improvements in empirical coverage, while further increasing to $M = 200$ results in only marginal changes across all nominal levels. This indicates diminishing returns beyond $M \approx 100$ and suggests that uncertainty estimates are already stable at this ensemble size. These results support the use of $M = 100$ samples throughout the uncertainty evaluation as a balance between computational cost and calibration accuracy.

*Table 4.* Empirical coverage as a function of ensemble size $M$, averaged over 20 test days.

| Nominal level | $M = 50$ | $M = 100$ | $M = 200$ |
|---|---|---|---|
| 50% | 0.469 | 0.472 | 0.473 |
| 70% | 0.683 | 0.685 | 0.684 |
| 90% | 0.899 | 0.901 | 0.902 |
| 95% | 0.955 | 0.957 | 0.958 |

### B.3. Low-resolution consistency.

To assess whether the generated high-resolution outputs remain consistent with the conditioning input, we downsample PODiff reconstructions back to the ACCESS-S2 grid and compare them with the original low-resolution input over all 2011 test days. PODiff achieves a coarse-grid RMSE of 0.187, $R^2$ of 0.948, and SSIM of 0.9815. This indicates that the POD-space conditioning preserves the low-resolution input structure in practice, despite not explicitly imposing a separate observation-matching term during sampling.

*Table 5.* Low-resolution consistency after downsampling PODiff outputs back to the ACCESS-S2 grid.

| Method | RMSE $\downarrow$ | $R^2 \uparrow$ | SSIM $\uparrow$ |
|---|---|---|---|
| PODiff-K40 | 0.187 | 0.948 | 0.9815 |

## C. Advection–Diffusion Reliability Curves

This appendix evaluates PODiff on a controlled advection–diffusion problem to complement the SST experiments and assess uncertainty behavior in a simplified setting with known dynamics.

Figure 8 presents reliability curves for PODiff in the controlled advection–diffusion test case. The close agreement between empirical and nominal coverage indicates well-calibrated predictive uncertainty. The tight clustering of individual realizations around the mean suggests that calibration behavior is consistent across test snapshots rather than driven by a small number of outliers.

These results mirror the uncertainty behavior observed in the SST experiments and demonstrate that PODiff maintains stable calibration properties in a simplified PDE setting with known dynamics.

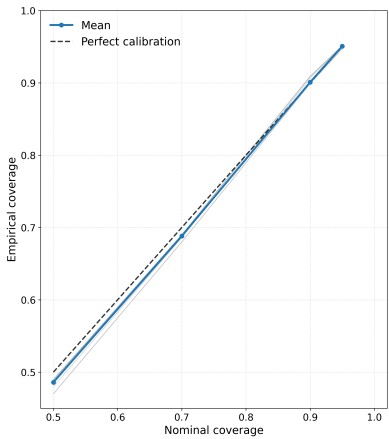

*Figure 8.* Reliability curves for PODiff on the advection–diffusion test case. The solid line shows mean empirical coverage across test snapshots, while faint lines indicate individual realizations.

## C.1. Advection-dominated regime

To evaluate behavior when the POD spectrum decays more slowly, we additionally consider an advection-dominated setting with small diffusivity and large Peclet number. In this regime, the 99% POD energy threshold requires $K = 150$ modes, compared with $K = 40$ in the smoother advection–diffusion benchmark. PODiff remains substantially smaller than PixelDiff, but the efficiency gap narrows as the intrinsic dimensionality increases.

*Table 6.* Advection-dominated PDE results. PODiff remains efficient when the number of retained POD modes increases.

| Method | $K$ | RMSE $\downarrow$ | MAE $\downarrow$ | Params |
|---|---|---|---|---|
| PODiff | 150 | 0.0291 | 0.0135 | 1.76M |
| PixelDiff | – | 0.0311 | 0.0140 | 33M |

For $M = 100$ ensemble samples, PODiff requires approximately 45s compared with 124s for PixelDiff. These results indicate that PODiff's efficiency advantage is largest for rapidly decaying spectra, but remains meaningful in more advection-dominated settings.

