# OpenReview forum: "PODiff: Latent Diffusion in Proper Orthogonal Decomposition Space for Scientific Super-Resolution"
_ICML.cc/2026/Conference — ICML 2026 regular_

### Official Review · Reviewer_3VBc · 2026-03-12

**Soundness:** 2
**Presentation:** 3
**Significance:** 2
**Originality:** 2
**Overall Recommendation:** 3
**Confidence:** 4

**Summary:**

This paper proposes a POD-based latent diffusion framework for super-resolution, where POD is used to construct latent variables and diffusion is performed in that latent space. The method is evaluated on real sea surface temperature data and a synthetic advection-diffusion benchmark, with comparisons against a pixel-space diffusion model (PixelDiff). The main empirical result is that the proposed method achieves accuracy comparable to PixelDiff at substantially lower computational cost, while also allowing second-order uncertainty estimates to be mapped analytically from latent space to physical space.

**Compliance With Llm Reviewing Policy:**

Affirmed.

**Final Justification:**

The paper presents a computationally efficient POD-based latent diffusion approach with analytically tractable uncertainty propagation, and the rebuttal strengthens the empirical case through the added VAE-based latent diffusion baseline. My two remaining concerns are that the paper still does not sufficiently explain why the POD latent geometry should be especially suitable for non-Gaussian diffusion-based generation, beyond providing an efficient linear representation with tractable second-order statistics, and that the practically important multivariable setting remains future work rather than a demonstrated capability. I therefore revise my overall recommendation to Weak Reject.

**Key Questions For Authors:**

N/A

**Limitations:**

yes

**Strengths And Weaknesses:**

**Strengths**
- The method avoids training an additional encoder-decoder for latent representation.
- It achieves accuracy comparable to PixelDiff at substantially lower computational cost.
- The linear reconstruction makes uncertainty propagation (from latent to physical space) analytically tractable.

**Weaknesses**
- The methodological novelty is limited: the paper mainly combines POD-based dimensionality reduction with a standard conditional diffusion, and the specific benefits of this combination are not yet fully established.
- A central claim that POD provides a suitable latent space for diffusion (L173-176) is not sufficiently supported. In particular, there is no comparison with learned latent representations, such as VAE-based latent diffusion. At a minimum, such a baseline should be included to support the claim that POD offers an advantage beyond computational convenience.
- The interpretability claim is somewhat overstated. The benefit appears mainly descriptive and limited to second-order statistics (L582-588), rather than providing deeper insight into non-Gaussian generative modeling.
- The method is only demonstrated on single-variable fields, and it remains unclear how it would extend to multivariate settings with different units, scales, and cross-variable correlations.
- The paper provides substantial implementation details, but it does not mention code release or code availability.

---

> ### Author Rebuttal · Authors · 2026-03-29
>
> We thank the reviewer for the detailed feedback, and for recognizing the computational efficiency and uncertainty propagation advantages of the approach. We address the main concerns below.
>
> ## **Novelty and contribution**
>
> The novelty claim is not about the components individually, but that (a) the orthogonality of POD makes the covariance propagation exact and training-free, a property no VAE-based LDM can offer, and (b) the variance-ordering creates a structured geometry with direct physical meaning. We agree these are not new generative paradigms, but the implications for scientific UQ are the contribution.
>
> We will revise the manuscript to make this point explicit rather than presenting Eq. 3 incidentally.
>
> ## **Comparison with learned latent diffusion (VAE baseline)**
>
> We have now included a VAE-based latent diffusion baseline. We trained a standard latent diffusion model (VAE-based, following Rombach et al., 2022) on the same SST task. The encoder-decoder (U-Net, C=64) compresses fields to an 8x8x40 spatial latent map (2560 dims), with a U-Net diffusion model (C=128, matching PixelDiff) in the learned latent space.
>
> **Results:**
>
> - RMSE: PODiff: **0.3923** vs VAE-LDM: 0.4011
> - CRPS: **0.2889** vs 0.2915
> - 90% coverage: 0.9009 vs 0.9022
>
> Despite similar accuracy and calibration, PODiff uses a **64x smaller latent space** (40 coefficients vs 2560 features), requires no encoder–decoder training, and provides closed-form uncertainty propagation via the POD basis. Moreover, the VAE-LDM requires ~39M total parameters and 28h two-stage training, compared to PODiff's 0.20M and 3.8h. This shows that PODiff achieves competitive or better performance without relying on learned latent representations.
>
> We will add this comparison to the revision.
>
> ## **Interpretability clarification**
>
> In operational settings (e.g., ensemble ocean forecasting), UQ primarily requires spatially resolved, well-calibrated predictive intervals for risk-aware decisions. PODiff’s analytic second-order structure directly supports this. We agree this is descriptive rather than mechanistic and will revise the manuscript to reflect this scope precisely.
>
> ## **Out-of-domain generalization**
> We applied PODiff to chest X-ray super-resolution (4x), comparing directly against PixelDiff on identical data:
>
> - FID: PODiff **9.65** vs PixelDiff 9.80
> - 90% coverage: PODiff **0.9008** vs PixelDiff 0.9016
> - 95% coverage: PODiff **0.9510** vs PixelDiff 0.9502
>
> PODiff matches PixelDiff on both reconstruction quality and calibration in a structurally different modality. This demonstrates that the analytic UQ mechanism generalises beyond geophysics, while retaining the full computational advantage (19x fewer parameters, 13x faster training).
>
> ## **Single-variable limitation**
>
> We acknowledge this limitation. POD can naturally extend to multi-variable fields (e.g., concatenating SST, salinity, and velocity components before computing the joint basis), and the framework presented here can apply without architectural changes. However, handling variables with different units, scales, and physical correlation structures requires careful normalization and validation that we have not yet performed. We will discuss this explicitly as a future direction.
>
> ## **Code availability and reproducibility**
>
> All implementation details required for reproducibility are provided in the paper. We will release the full codebase (training, inference, and evaluation) upon acceptance. We will add a explicit statement to this effect in the revised manuscript.
>
> ## **Summary**
>
> The VAE baseline directly addresses the central concern about POD's suitability as a latent space. The cross-domain result demonstrates generalization. We have clarified the scope of interpretability and acknowledged the multivariate limitation explicitly. We respectfully ask the reviewer to reconsider their score in light of these additions.

---

> > ### Author Rebuttal · Reviewer_3VBc · 2026-04-03
> >
> > The rebuttal strengthens the empirical case, and I appreciate the added VAE-based latent diffusion baseline. However, I would still have appreciated a clearer explanation of what is meant by “the variance-ordering creates a structured geometry with direct physical meaning,” especially for non-Gaussian generation rather than second-order structure alone. Since POD is fundamentally a linear change of basis with truncation, I am still not fully convinced that the paper explains why this latent-space geometry is especially well suited for diffusion-based generation, or how it contributes to the observed accuracy gains. For this reason, I remain unsure whether the reported advantage reflects a broader property of the method or is mainly specific to the examples considered here. I will therefore revise my recommendation to Weak Reject.

---

> > > ### Author Response · Authors · 2026-04-04
> > >
> > > Thank you for reconsidering and for the thoughtful follow-up. While POD is a linear basis, its variance-ordering creates a latent space where dimensions are ranked by energy contribution, with orthogonal modes corresponding to descending covariance eigenvalues. For diffusion models, this induces a natural coarse-to-fine denoising process: early timesteps operate on dominant modes capturing large-scale structure, while later timesteps refine lower-energy details. This behavior arises from the latent geometry itself and is not tied to Gaussian assumptions, but reflects the ordering of modes by energy, which we observe consistently across datasets, including in our cross-domain experiments (e.g., X-ray super-resolution). This alignment leads to more structured sampling, efficient representation, and facilitates uncertainty propagation compared to unstructured latent spaces. We will revise the manuscript to make this connection explicit.

---

### Official Review · Reviewer_WcPG · 2026-03-12

**Soundness:** 3
**Presentation:** 3
**Significance:** 3
**Originality:** 3
**Overall Recommendation:** 4
**Confidence:** 3

**Summary:**

The paper proposes a probabilistic super-resolution framework, PODiff, that performs conditional diffusion in a fixed, variance-ordered latent space defined by Proper Orthogonal Decomposition (POD). Instead of operating in pixel space, it diffuses POD coefficients and reconstructs fields via linear mapping, enabling efficient ensemble generation and analytic uncertainty propagation. Experiments on sea surface temperature downscaling (640×480) and a controlled advection-diffusion problem show PODiff achieves accuracy comparable to pixel-space diffusion while reducing computational cost and producing well-calibrated, spatially interpretable uncertainties.

**Compliance With Llm Reviewing Policy:**

Affirmed.

**Final Justification:**

The authors's rebuttal has addressed my concerns. I'll maintain my positive score.

**Key Questions For Authors:**

1) Is there a practical heuristic for selecting beyond the 99% variance threshold?
2) For fields that are not low-rank (e.g., turbulence), how many modes would be needed, and at what point does PODiff become computationally impractical compared to pixel diffusion?

**Limitations:**

The proposed method relies on low-rank linear structure; performance may degrade for highly turbulent or discontinuous fields. This is acknowledged but not explored.

**Strengths And Weaknesses:**

Strengths:
1) The paper include thorough comparison against multiple baselines on real-world SST data and a controlled PDE benchmark.
2) Linear reconstruction enables analytic covariance propagation and mode-level inspection, a clear advantage over black-box latent spaces

Weaknesses:
1) The paper didn't include comparison to learned latent diffusion (e.g., VAE-based), but including such a comparison would strengthen claims about POD's advantages for scientific data.

---

> ### Author Rebuttal · Authors · 2026-03-29
>
> We thank the reviewer for the positive assessment and for highlighting the advantages of analytic uncertainty propagation and computational efficiency. We address the main points below.
>
> ## **Comparison with learned latent diffusion**
>
> We agree that comparison with learned latent diffusion is important and have now implemented a VAE-based latent diffusion model (following Rombach et al., 2022) on the same SST task. The encoder-decoder (U-Net, C=64) compresses fields to an 8x8x40 spatial latent map (2560 dims), with a U-Net diffusion model (C=128, matching PixelDiff) in the learned latent space.
>
> **Results:**
>
> - RMSE: PODiff: **0.3923** vs VAE-LDM: 0.4011
> - CRPS: **0.2889** vs 0.2915
> - 90% coverage: 0.9009 vs 0.9022
>
> PODiff achieves competitive or better performance using a 64x smaller latent space (40 flat coefficients vs 2560 spatial features), with no encoder-decoder training. Moreover, the VAE-LDM requires ~39M total parameters and 28h two-stage training, compared to PODiff's 0.20M and 3.8h. This directly addresses the reviewer's question: in this setting, the encoder–decoder cost offsets the potential parameter savings of a learned latent space.
>
> We will add this comparison to the revision.
>
> ## **Heuristic for selecting variance threshold**
>
> The 99% cumulative variance threshold is a standard heuristic in the reduced-order modeling literature and worked well in practice for SST. Moreover, we validated the choice of $K$ empirically by varying $K \in \{10, 20, 40\}$ as shown in Tab. 1. In practice, we recommend a two-step approach: (1) select $K$ via the variance threshold as an initial estimate, then (2) validate on held-out data by monitoring reconstruction error and diffusion validation loss as a function of $K$. If validation loss plateaus, adding more modes provides diminishing benefit.
>
> ## **Behavior for non–low-rank fields**
>
> This is an important question. The computational advantage of PODiff comes from operating an MLP on $K$ coefficients rather than a U-Net on the full spatial grid. As $K$ increases for fields with slower spectral decay, the MLP grows. However, the important thing here is that even at K=200, the MLP would remain lightweight, on the order of a few million parameters. On the other hand, PixelDiff has 33M parameters operating on 640x480 spatial fields (See Tab. 3). The crossover point where PODiff loses its efficiency advantage is therefore relatively high in practice.
>
> ## **Advection-dominated problems**
> We ran an additional experiment specifically targeting an advection-dominated regime, where $\nu \sim [1e-5,1e-4]$ and Péclet number reaches $O(10^6)$.
>
> **SVD decay:** The spectrum decay slows and K=150 modes are needed to capture the 99% variance.
>
> **PODiff at K=150:** PODiff achieves **RMSE=0.0291, MAE=0.0135**, using only **1.76M parameters vs PixelDiff's 33M (19x fewer)**, with M=100 ensemble taking **~45s vs 124s (2.8x faster)**, while retaining closed-form UQ.
>
> We agree that more POD modes may be needed for highly turbulent or discontinuous fields where singular value decay is even more slower. This is a limitation of linear reduced-order representations rather than diffusion itself. Adaptive basis construction is a promising future direction to solve this issue.
>
> ## **Out-of-domain generalization**
> To demonstrate generality beyond geophysical data, we applied PODiff to chest X-ray super-resolution (4x), achieving FID = 9.65, indicating the framework extends to different modalities.
>
> ## **Summary**
>
> With the addition of the VAE-based latent diffusion baseline addressing the comparison with learned latent spaces, and further clarification of PODiff’s behavior in more challenging regimes, we believe the key concern has been addressed. We hope these additions strengthen the empirical support for the method and respectfully ask the reviewer to reconsider their score.

---

> > ### Author Rebuttal · Reviewer_WcPG · 2026-04-02
> >
> > The authors's rebuttal has addressed my concerns. I'll maintain my positive score.

---

> > > ### Author Response · Authors · 2026-04-02
> > >
> > > Thank you for acknowledging the rebuttal and for the positive assessment. We are glad that the VAE-LDM comparison and the additional experiments on advection-dominated regimes addressed your concerns satisfactorily.
> > >
> > > Given that the main weakness you identified has now been addressed with the added baseline and supporting analysis, we hope you may consider raising your score accordingly. We leave this to your discretion and greatly appreciate your thoughtful and constructive engagement with our work.

---

### Official Review · Reviewer_JCLC · 2026-03-13

**Soundness:** 3
**Presentation:** 3
**Significance:** 2
**Originality:** 3
**Overall Recommendation:** 4
**Confidence:** 3

**Summary:**

PODiff presents a technique for scientific super-resolution using diffusion models by working in a latent space induced by the Proper Orthogonal Distribution. The approach is motivated by connections to reduced-order modeling and the challenges of diffusion model application in pixel space for high-dimensional data. The technique is novel and the paper is well-written.

**Compliance With Llm Reviewing Policy:**

Affirmed.

**Final Justification:**

The rebuttal adequately addressed my concerns and made me change my evaluation towards a weak accept. Precise numbers indicating speedup and storage reduction influenced my decision.

**Key Questions For Authors:**

1. In Table 1, it appears that PixelDiff is almost as good as PODiff-K40, and that PODiff-K20 and PODiff-K10 struggle in comparison. What is the training cost of PixelDiff versus PODiff-K40? Is there a cost of constructing the POD basis in PODiff-K40? Is the performance gain overcoming the computational cost?
2. PixelDiff appears extremely close to PODiff-K40. Is there a justification for using PODiff over PixelDiff in general?
3. Should the POD approach also limit spatial resolution due to smoothing induced by truncating the SVD?
4. What is the ratio of d_{low} against d (e.g., 4 X or 8 X)?

**Limitations:**

Beyond not addressing advection-dominated problems and more complicated PDEs, there are no concerns about inadequately discussed limitations.

**Strengths And Weaknesses:**

There are concerns about whether the results are competitive against existing techniques, as well as whether the technique will scale to advection-dominated problems. There are also several technical questions about the methods being used as baselines. Being clear about the limitations of this approach would be highly useful.

It is also unclear whether a POD-based approach would work for an advection-dominated problem. The utility of this approach for more complicated PDEs should be discussed and investigated.

Some minor presentation issues are as follows:

1. The best-performing techniques should be boldfaced to make it easier to understand Table 1.
2. I assume lower values are better for the results in Table 1 and higher values are better for the results in Table 2. Arrows should be present to make this easier to understand.
3. Consistency in paragraph titles (e.g., on page 5, we have “High resolution data” followed by “Low-resolution inputs.”).

Other issues are as follows:

1. The advection-diffusion problem in Section 4.3 needs to be discussed in more detail. The PDE, coefficients, and boundary conditions are not described.
2. What is the kernel width for the RBF baseline and how is it chosen? Do we know whether it is optimal or not?
3. Why is bicubic interpolation used in the low-resolution inputs but the RBF approach is used as the baseline?

Demonstrating PODiff for advection-dominated tasks would be highly convincing and insightful.

---

> ### Author Rebuttal · Authors · 2026-03-29
>
> We thank the reviewer for the thoughtful and constructive feedback. We address the main concerns below and will add all clarifications in the final version.
>
> ## **Questions 1 and 2**
>
> ### **(a) PixelDiff vs PODiff**
>
> PODiff offers clear advantages over PixelDiff through 15x lower operational cost, closed-form uncertainty propagation, and strong cross-domain generalization beyond comparable SST accuracy.
>
> **1. Operational efficiency:** For ensemble forecasting (M = 100 samples), PixelDiff requires 124s per case vs PODiff’s 8s, which over one year of daily forecasting (365 × 100 samples) translates to ~12.6 hours vs ~49 minutes (15x reduction). This is not a marginal gain, but a difference that determines whether ensemble downscaling is practically deployable.
>
> **2. Computational efficiency (Tab. 3).**
> - **165x fewer parameters** (0.20M vs 33M)
> - **~9x lower GPU memory** (1.4 vs 12.5 GB)
> - **~13x faster training** (3.8 vs 48 hours)
> - **~15x faster sampling** (0.08 vs 1.24 s per sample).
>
> Additionally, due to linear reconstruction (Eq. 3), PODiff enables direct propagation of latent uncertainty to spatial covariance, yielding structured and interpretable uncertainty not available in pixel-space diffusion.
>
> **3. Comparison with learned latent diffusion (new result).**
> We trained a standard latent diffusion model (VAE-based, following Rombach et al., 2022) on the same SST task. The encoder-decoder (U-Net, C=64) compresses fields to an 8x8x40 spatial latent map (2560 dims), with a U-Net diffusion model (C=128) in the learned latent space.
>
> - PODiff-K40 RMSE: **0.3923** vs VAE-LDM: 0.4011
> - CRPS: **0.2889** vs 0.2915
> - 90% coverage: 0.9009 vs 0.9022
>
> PODiff uses a **64x smaller latent space** (40 coefficients vs 2560 features), requires no encoder–decoder training, and provides closed-form uncertainty propagation via the POD basis. Moreover, the VAE-LDM requires ~39M total parameters and 28h two-stage training, compared to PODiff's 0.20M and 3.8h. This shows that PODiff achieves comparable performance without relying on learned latent representations.
>
> ### **(b) Cost of POD basis construction**
>
> The POD basis is computed once via SVD on training data that takes approximately 1 minute. Negligible relative to the 3.8h diffusion training cost.
>
> ## **POD truncation and spatial resolution (Question 3)**
>
> Truncation does impose a resolution floor: the discarded modes (beyond $K=40$) account for $\textless$ 1% of total variance, bounding the maximum reconstruction error attributable to truncation. In practice, sharp gradients are preserved at this level (Fig. 2), and the diffusion model further corrects residual errors beyond the deterministic POD projection (RMSE 0.3923 vs 0.7084), indicating the model learns to recover structure not captured by the projection alone. Lower truncations ($K = 10, 20$) show clear degradation (Tab. 1), confirming that $K$ directly controls the resolution-efficiency trade-off.
>
> ## **Ratio of d_{low} to d (Question 4)**
>
> SST: 53x31 → 640x480, **~12x spatial upscaling** (pixel ratio ~187x). Advection–diffusion: 32x32 → 128x128, **4x upscaling** (pixel ratio 16x). We will clarify this in the revised paper.
>
> ## **Advection-dominated problems**
>
> We ran an additional experiment specifically targeting an advection-dominated regime, where $\nu \sim [1e-5,1e-4]$ and Péclet number reaches $O(10^6)$. The spectrum decay slows and K=150 modes are needed to capture the 99% variance.
>
> **PODiff at K=150:** PODiff achieves **RMSE=0.0291, MAE=0.0135**, using only **1.76M parameters vs PixelDiff's 33M (19x fewer)**, with M=100 ensemble taking **~45s vs 124s (2.8x faster)**, while retaining closed-form UQ.
>
> The efficiency advantage scales with the problem's intrinsic dimensionality. It is maximal for fast-decaying spectra (e.g., SST, smooth PDEs), and diminishing but meaningful for strongly advection-dominated fields. We will add SVD decay curves across regimes to the appendix. The Péclet number and full PDE description will be stated prominently in the revision.
>
> ## **RBF baseline and interpolation**
>
> Bicubic interpolation is used only for grid alignment. RBF uses a thin-plate spline kernel (parameter-free). We will clarify this distinction.
>
> ## **Out-of-domain generalization**
> We applied PODiff to chest X-ray super-resolution (4x), with direct PixelDiff comparison on identical data:
> * FID: PODiff **9.65** vs PixelDiff 9.80
> * 90% coverage: PODiff **0.9008** vs PixelDiff 0.9016
> * 95% coverage: PODiff **0.9510** vs PixelDiff 0.9502
>
> PODiff matches PixelDiff and retains the full computational advantage (19x fewer parameters, 13x faster training).
>
> ## **Summary**
> The advection-dominated experiment directly addresses the primary concern. The VAE baseline contextualizes PODiff against learned latent alternatives. The operational efficiency numbers quantify why parity accuracy at 15x lower ensemble cost justifies the approach. We respectfully ask the reviewer to reconsider their score.

---

> > ### Author Rebuttal · Reviewer_JCLC · 2026-04-03
> >
> > I thank the authors for their detailed response! The careful and detailed rebuttal has made me adjust my score towards a weak accept. I am still concerned about general advection-dominated problems and wish to see the precise PDE setup, but am sufficiently convinced by the additional results provided. I wish the authors the best of luck in the review process!

---

> > > ### Author Response · Authors · 2026-04-04
> > >
> > > Thank you for the thoughtful follow-up and for revising your assessment. We are glad that the additional experiments and clarifications addressed your concerns. We also appreciate your suggestion regarding the precise PDE setup for advection-dominated regimes, and will include a clearer description in the final version. Thank you again for your constructive feedback throughout the review process.

---

### Official Review · Reviewer_QXCQ · 2026-03-13

**Soundness:** 3
**Presentation:** 3
**Significance:** 2
**Originality:** 2
**Overall Recommendation:** 4
**Confidence:** 4

**Summary:**

This paper proposes conditional diffusion (specifically for scientific super-resolution) using an orthogonal decomposition space instead of the traditional pixel space or learned latent space. Unlike images, scientific data is often more easily represented with low-rank structure, and may also be much higher resolution, causing scaling issues with traditional super-resolution methods. The low res input is upsampled, then converted to bases components. That is used as a conditioning signal for the diffucion model which then outputs the parameters for a high-res output, which is then reconstructed via the orthogonal bases.

**Compliance With Llm Reviewing Policy:**

Affirmed.

**Key Questions For Authors:**

Can you add a direct comparison to a learned latent diffusion baseline or is the complexity of doing that too high?

How closely do the outputs match the inputs when downsampled back to the same input space?

**Limitations:**

Yes, the authors discuss limitations, in a way that is mostly comprehensive. They note the dependence on low-rank linear structure, potential degradation for turbulent/discontinuous fields, sensitivity to distribution shift, and the fact that truncation uncertainty is not modeled.

**Strengths And Weaknesses:**

Strengths:
Although the method is straightforward, it is elegant in it's approach of using an orthognal basis instead of a learned latent space. The benefits of that especially regarding computation are clearly shown compared to a latent space. This is a sensible idea for scientific fields with strong low-rank structure, and the paper articulates the rationale clearly. The experiments are good. Specifically the inclusion of deterministic POD projection, random orthonormal latent diffusion, deterministic U-Net, MC Dropout U-Net, RBF interpolation, and pixel-space diffusion helps isolate what is gained from the specific POD structure versus diffusion versus standard supervised learning. Table 3 reports a very large reduction in parameters, GPU memory, and training time relative to PixelDiff, while maintaining similar reconstruction accuracy on SST.

Weakness:
The biggest weakness is that the novelty is somewhat moderate. The paper’s core move is to replace pixel-space or learned-latent diffusion with diffusion in a classical linear POD basis. That is a neat and useful combination, but it is closer to a well-motivated reparameterization plus careful application than to a fundamentally new generative modeling method. The paper does acknowledge that reduced-order representations and latent diffusion are both established directions, which is appreciated. But I do wonder if a learned latent space with fewer parameters would do just as well computationally and performance wise.

Additionally, compared to inverse diffusion methods that include a observation matching term, the condition vector is rather weak. This means that we have no guarantee that the super-res output would downsample and match the input. Additionally such reconstruction/input consistency numbers are not shown, only MAE and RMSE against the ground truth. Also the paper does not compare against an autoencoder latent diffusion baseline, which is the most obvious neighboring method class.

---

> ### Author Rebuttal · Authors · 2026-03-29
>
> We thank the reviewer for the thoughtful feedback and for recognizing the efficiency and elegance of the proposed approach. We address their main concerns below.
>
> ## **Comparison with learned latent diffusion**
>
> We have now included a VAE-based latent diffusion baseline. Following Rombach et al. (2022), we use a convolutional encoder-decoder (U-Net, C=64) that compresses fields to an 8x8x40 spatial latent map (2560 dims), with a U-Net diffusion model (C=128, matching PixelDiff) operating in the learned latent space.
>
> **Results:**
>
> - RMSE: PODiff: **0.3923** vs VAE-LDM: 0.4011
> - CRPS: **0.2889** vs 0.2915
> - 90% coverage: 0.9009 vs 0.9022
>
> PODiff achieves competitive or better performance using a 64x smaller latent space (40 flat coefficients vs 2560 spatial features), with no encoder-decoder training. Moreover, the VAE-LDM requires ~39M total parameters and 28h two-stage training, compared to PODiff's 0.20M and 3.8h. This directly addresses the reviewer's question: in this setting, the encoder–decoder cost offsets the potential parameter savings of a learned latent space.
>
> We will add this comparison to the revision.
>
> ## **Conditioning strength and reconstruction consistency**
>
> We thank the reviewer for this important point. We evaluated coarse-grid consistency by downsampling PODiff's high-resolution output back to the ACCESS-S2 low-resolution grid and comparing against the original low-resolution input. For the test set consisting of all days in the calendar year 2011, we got an RMSE of 0.187, $R^2$ of 0.948 and SSIM of 0.9815. These results indicate that PODiff's super-resolution outputs are highly consistent with the original low-resolution inputs when projected back to the coarse grid.
>
> We note that the conditioning signal is provided via projection into the POD basis, which enforces structural alignment between low and high-resolution representations. The high $R^2$ and SSIM values above confirm that this implicit conditioning is effective in practice. We will add this consistency analysis to the revision.
>
> ## **Novelty clarification**
>
> We agree that the core idea can be viewed as a structured reparameterization. We would like to clarify that the contribution lies in:
>
> * Analytic uncertainty propagation from latent to physical space (Eq. 3):  It relies on the orthogonality and linearity of the POD reconstruction. This property does not generally hold for learned latent diffusion models.
> * Variance-ordered latent geometry: The POD basis provides a fixed, hierarchical organization of spatial scales. This is structurally distinct from learned latent spaces, where the latent geometry is unconstrained and mode-level inspection is not possible. This property is specifically valuable in scientific ML applications (e.g., climate science, oceanography), where researchers routinely analyze dominant spatial patterns through empirical orthogonal functions.
> * Parameter-efficient probabilistic modeling: PODiff works without encoder–decoder training.
>
> We will revise the manuscript to clarify that the novelty lies in this **integration and its implications (efficiency + structured uncertainty)**, rather than in introducing a fundamentally new generative paradigm.
>
> ## **Out-of-domain generalization**
> To demonstrate generality beyond geophysical data, we applied PODiff to chest X-ray super-resolution (4x), achieving FID = 9.65, indicating the framework extends to different modalities.
>
> ## **Summary**
>
> The VAE-LDM comparison directly addresses the reviewer's key question and confirms that POD provides an efficient and structured latent space. The consistency analysis also demonstrates that PODiff outputs faithfully preserve low-resolution input structure. We hope these additions strengthen the reviewer's assessment.

---

> > ### Author Rebuttal · Reviewer_QXCQ · 2026-04-08
> >
> > The inclusion of a VAE-based latent diffusion baseline directly addresses one of my main concerns. The results suggest that PODiff achieves comparable or slightly better performance while being significantly more parameter- and compute-efficient. This strengthens the paper’s empirical case and clarifies that the benefits are not solely due to operating in a lower-dimensional space, but also due to the structure of the POD representation.
> >
> > I also appreciate the reconstruction evaluation numbers. This addresses my concern about conditioning strength in practice.
> >
> > Overall, the rebuttal strengthens the paper and resolves my main questions. However, my overall assessment remains unchanged. While the additional results improve confidence in the approach, it does not substantially change my view of the paper’s overall novelty and scope of impact.

---

> > > ### Author Response · Authors · 2026-04-08
> > >
> > > Thank you for the thoughtful follow-up and for carefully considering the rebuttal. We are glad that the additional experiments, particularly the VAE-based latent diffusion comparison and reconstruction evaluation, helped clarify the empirical behavior and address the main concerns.
> > >
> > > We appreciate your assessment regarding the role of the POD representation. Our intention is to highlight that the structured, variance-ordered latent space enables not only efficiency but also analytically grounded and spatially interpretable uncertainty, which we view as a key contribution of the work.
> > >
> > > Thank you again for your constructive feedback.

---

### Decision · Program_Chairs · 2026-04-30

**Decision:**

Accept (regular)

**Comment:**

Reviewers agree that this paper represents a sound and well-motivated approach to super-resolution, and appreciate the computational benefits of the approach while maintaining strong performance. However, there are some concerns regarding the novelty of the approach, as well as some concerns regarding the choice of baselines. The latter was partially addressed with preliminary findings during the rebuttal period. Overall, this paper represents a practical contribution for scientific ML, though with somewhat limited novelty and scope.